# Removal of Protein-Bound Uremic Toxins Using Binding Competitors in Hemodialysis: A Narrative Review

**DOI:** 10.3390/toxins13090622

**Published:** 2021-09-04

**Authors:** Vaibhav Maheshwari, Xia Tao, Stephan Thijssen, Peter Kotanko

**Affiliations:** 1Renal Research Institute, New York, NY 10065, USA; Xia.Tao@rriny.com (X.T.); Stephan.Thijssen@rriny.com (S.T.); Peter.Kotanko@rriny.com (P.K.); 2Icahn School of Medicine at Mount Sinai, New York, NY 10029, USA

**Keywords:** binding competition, hemodialysis, intoxication, indoxyl sulfate, p-cresyl sulfate, CMPF

## Abstract

Removal of protein-bound uremic toxins (PBUTs) during conventional dialysis is insufficient. PBUTs are associated with comorbidities and mortality in dialysis patients. Albumin is the primary carrier for PBUTs and only a small free fraction of PBUTs are dialyzable. In the past, we proposed a novel method where a binding competitor is infused upstream of a dialyzer into an extracorporeal circuit. The competitor competes with PBUTs for their binding sites on albumin and increases the free PBUT fraction. Essentially, binding competitor-augmented hemodialysis is a reactive membrane separation technique and is a paradigm shift from conventional dialysis therapies. The proposed method has been tested in silico, ex vivo, and in vivo, and has proven to be very effective in all scenarios. In an ex vivo study and a proof-of-concept clinical study with 18 patients, ibuprofen was used as a binding competitor; however, chronic ibuprofen infusion may affect residual kidney function. Binding competition with free fatty acids significantly improved PBUT removal in pre-clinical rat models. Based on in silico analysis, tryptophan can also be used as a binding competitor; importantly, fatty acids or tryptophan may have salutary effects in HD patients. More chemoinformatics research, pre-clinical, and clinical studies are required to identify ideal binding competitors before routine clinical use.

## 1. Introduction

Uremic toxins have broadly been classified into three categories: (1) small-sized toxins (<500 Da), (2) middle and large-sized uremic toxins (>500 Da), and (3) protein-bound uremic toxins (PBUTs) [1]. In an updated definition, these solutes were further classified based on their origins, molecule weight, and albumin-binding properties [2]. In end-stage kidney disease (ESKD) patients, these toxins accumulate, and patients must undergo dialysis to reduce the levels of these toxins. Conventional hemodialysis (three times a week, four hours per session) provides adequate removal of small, non-protein-bound solutes such as urea and creatinine. Removal of middle-sized toxins such as β_2_-microgloblin is also improved by convection-based hemodiafiltration (HDF). However, removal of PBUTs is particularly poor during conventional hemodialysis (HD) [3], and HDF provides only marginal improvement over HD [4,5]. PBUT removal in the extracorporeal renal replacement therapies is poor because the majority of PBUTs are tightly bound to protein, with albumin being the primary carrier protein, and only a small free fraction is available for transfer across the dialyzer membrane [6]. The lower free toxin concentration results in a smaller diffusive gradient for toxin removal in dialysis, while convection is not helpful due to membrane size cut-off to retain proteins and, hence, albumin-bound toxins. 

For the prototypical PBUTs indoxyl sulfate (IS) and p-cresyl sulfate (pCS), with known toxicities in chronic kidney disease patients [2,7], the albumin-bound fraction is approximately 95% [3]. The poor removal of PBUTs with conventional HD is evidenced by the fact that pre-dialysis levels of IS and pCS have been found to be as much as 116-fold and 41-fold higher, respectively, than in age-matched healthy controls, while pre-dialysis concentrations of non-protein-bound solutes with comparable molecular weight, such as urea and creatinine, were only 5- and 13-fold higher, respectively [8]. Strong protein-binding results in a low reduction ratio of 20 to 35% for prototypical PBUTs [9], while the reduction ratio for small non-protein-bound solutes such as urea is around 80% during a conventional 4 h hemodialysis session [10]. The reduction ratio of a solute is a measure of dialysis session efficacy and indicates the percentage reduction in total serum concentration for this solute during a given dialysis duration. For PBUTs such as 3-carboxy-4-methyl-5-propyl-2-furanpropionate (CMPF), very strong binding to albumin results in a negligible free fraction, such that their removal is practically zero. The reduction ratio for CMPF with conventional HD is often negative [3,11]. Note that the negative reduction ratio of CMPF does not suggest that there is an increase in CMPF generation rate due to dialysis; rather, it is an indication of the fact that protein concentration (and thus bound CMPF) is increased due to ultrafiltration. 

To remove this class of toxin, we proposed a method where PBUTs can be removed by competitive binding, also known as the displacer method [12,13]. In the sections below, we will first describe the concept, followed by evidence ranging from bench to clinical studies. Subsequently, we summarize in silico evidence and comparisons of various extracorporeal modalities with respect to PBUT removal. We also highlight the potential application of this binding competition approach to treat drug intoxications. Finally, we provide a brief discussion of the other technologies that aim to improve PBUT removal and conclude with thoughts on the next steps to make this concept a clinical reality.

## 2. Binding Competition for PBUT Removal

Binding competition is a well-known and often-utilized process in pharmacology. Fundamentally, much of the occupancy-driven drug pharmacology acts on the premise of binding to a receptor/protein of interest. An administered drug molecule can work as a receptor agonist, antagonist, or inhibitor, while an endogenous ligand molecule may also be competing for the same binding site on the receptor. Binding competition studies are also critical for elucidating potential drug–drug interactions, especially in patients with complex medication regimen due to multiple comorbidities. In the proposed concept, we utilized the same binding competition method but introduce the binding competitor into the extracorporeal circuit where it competes for the same binding site as the PBUTs on the albumin molecule (Figure 1). This renders a larger fraction of the PBUTs unbound, which increases the diffusion gradient and improves dialytic removal.

While terms like “PBUT displacement” and “displacer-enhanced dialysis” are occasionally used to refer to the concept, those are technically misnomers. Rather than displacement [illustrated in Equation (1) below], the mechanism at play is competitive binding [illustrated in Equation (2) below]. PBUTs are reversibly bound to albumin with weak Van der Waals forces and are always in a dynamic equilibrium with the carrier protein. When an exogenous compound that binds to the albumin PBUT binding site is introduced, it results in a reduced free protein concentration, leading to a shift in dynamic equilibrium to provide more free protein, as per Le Chatelier’s law of chemical equilibria [14] [see Equation (2)]. Thereby, in effect, more toxins also become free and thus dialyzable. These processes are amenable to quantitative analysis since protein–drug binding affinity data are available for most drug compounds [15].
(1)Equation 1: [Protein−toxin]+[Drug] ↔ [Protein−Drug]+[Toxin]
(2)Equation 2:[Protein−toxin] ↔ [Protein]+[Toxin]
[Protein−Drug] ↔ [Protein]+[Drug]

Though simple, binding competition significantly improves PBUT removal, as highlighted in the sections below.

## 3. Evidence from Bench Studies

In 2015, Tao et al. were the first to test the binding competition concept for enhancing the dialytic removal of PBUTs [13]. They used tryptophan (TRP) and docosahexaenoic acid (DHA) as IS competitors and validated the concept by measuring IS concentration in the dialysate outlet stream. In their in vitro setup, 4% human serum albumin (HSA) solution spiked with IS was circulated through the dialyzer. The binding competitors (TRP or DHA), infused on the blood side of the circuit upstream of the dialyzer, competed for the IS binding site on albumin and led to an increase in IS concentration in the dialysate outlet stream. The fractional removal of IS was expressed as the amount per unit of time leaving the dialysate outlet as a percentage of the amount per unit of time entering at the blood inlet. When infusing phosphate buffer solution (PBS) without any added binding competitor, IS removal was 10.2%, which improved to 18.5% with TRP and to 27.7% with DHA added to the infusion. Since the binding affinity of DHA to albumin (1.0 × 10^7^ M^−1^) is much higher than that of TRP to albumin (1.0 × 10^4^ M^−1^) [16], DHA is the stronger binding competitor of the two and leads to a larger improvement in dialytic removal of IS. Figure 2 shows a schematic of the bench setup used and summarizes the improvement in IS removal from baseline.

In 2016, Tao et al. further validated the binding competition approach ex vivo, where human whole blood spiked with IS, indole-3-acetic acid (IAA), and hippuric acid (HA) was dialyzed with either an ibuprofen + furosemide infusion or with a TRP infusion only, and toxin removal in the dialysate outlet stream was compared against phosphate-buffered saline infusion without binding competitors [17]. The ibuprofen and furosemide cocktail increased the IS fractional removal from 6.4% to 18.3%, and IAA fractional removal from 16.8% to 34.5%. TRP infusion increased the fractional removal of IS and IAA to 10.5% and 27.1%, respectively. Moderate effects were observed for HA in all infusion scenarios (Figure 3). This study confirmed important aspects about the binding affinity of toxins as well as that of binding competitors. For PBUTs with weak binding affinity to albumin, binding competitors provided only moderate gains in terms of toxin removal. Conversely, competitors with stronger binding affinity provided greater improvement in PBUT removal; here, the ibuprofen binding affinity (2.7 × 10^6^ M^−1^) was much higher than the TRP binding affinity [16]. Tao et al. did not use DHA in this ex vivo setup because it caused hemolysis (data not shown).

## 4. Evidence from Pre-Clinical Studies

In 2019, Li et al. used chronic kidney disease rat models and salvianolic acid infusion during microdialysis [18]. Salvianolic acids are phytochemicals and strongly bind to albumin. They are believed to have anti-oxidative properties. In 5/6 nephrectomized Sprague-Dawley rats on microdialysis, Li et al. collected a dialysate sample every 30 min for 4 h. The first 2 h of the experiment served as the control, while Danhong injection (DHI, mixture of salvianolic acids) was injected intravenously during the remaining 2 h of microdialysis. Comparing the infusion phase with the control, the IS and pCS removal improved by 135.6% and 272%, respectively. The authors also used lithospermic acid only (one of the salvianolic acids), which improved IS and pCS removal, respectively, by 119.5% and 127.5% in comparison to control. Higher removal by Danhong injection in comparison to lithospermic acid may be due to the cumulative effect of the number of salvianolic acids present in DHI. This study highlights an important point regarding the use of a binding competitor cocktail that targets multiple binding sites on albumin molecules to exert an even larger removal of PBUTs during dialysis.

In 2019, Shi et al. also validated the binding competition method in in vitro experiments and pre-clinical uremic rat models [19]. They first tested the inhibitory effect of free fatty acids (FFAs) on the albumin-binding of CMPF, pCS, IS, and IAA in an in vitro setup, where human albumin solution spiked with PBUTs was dialyzed against standard bicarbonate buffer. Infusion of FFAs upstream of the dialyzer increased the fractional removal of pCS, IS, and IAA from 8.00%, 11.68%, and 15.38%, respectively, at baseline to 28.21%, 35.42%, and 40.18%. CMPF fractional removal increased to 14.4%, with no removal at baseline. In the pre-clinical rat models, 16 weeks after 5/6 nephrectomy, a control group with saline infusion was compared to a treatment group with intralipid emulsion (ILE) infusion. Intravenous infusion occurred 30 min before the start of dialysis and allowed serum non esterified free fatty acid levels to reach six times higher than the control when dialysis started and remained elevated for most of the experiments. Total solute removal, measured using total dialysate collection in 180 min of dialysis, improved up to 300% for PBUTs in the ILE infusion arm, whereas it remained unchanged for the non-protein-bound solutes urea and creatinine (Figure 4). 

In a further pre-clinical study comprising a uremic rat model, Shi et al. 2021 studied albumin dialysis and binding competition separately, as well as in combination. They used 4% bovine serum albumin in bicarbonate dialysate for albumin dialysis and ω-6 soybean oil-based lipid emulsion as a binding competitor. Binding competition outperformed albumin dialysis and improved IS and pCS removal by approximately 10-fold in comparison to conventional 4 h HD. Notably, combining binding competition and albumin dialysis further improved the removal of putative PBUTs [20]. This study indirectly underscores the fact that protein binding is the primary resistance for PBUTs removal. Once this resistance is overcome by binding competition, combination therapies such as albumin dialysis or membrane adsorption with binding competition can significantly improve PBUTs removal. Note that the use of FFAs as binding competitors can not only improve PBUT removal but may also have salutary effects in dialysis patients with appropriate administration dosages.

## 5. Clinical Evidence

In 2019, Madero et al. were the first to study the effect of binding competition in a clinical proof-of-concept study with 18 patients [21]. In a 240 min conventional HD, they infused 800 mg ibuprofen into the arterial line of the extracorporeal circuit, upstream of the dialyzer, from minutes 21 to 40. They observed the dialysate clearances of IS, pCS, TRP, and non-protein bound solutes (Figure 5). Between the pre-infusion (0–20 min) and infusion periods (21–40 min), dialysate clearance of IS and pCS increased from 6.6 to 20 mL/min and 4.4 to 14.9 mL/min, respectively. TRP clearance increased moderately, while removal of urea and creatinine remained unchanged.

## 6. In Silico Evidence

To quantify total solute removal during a dialysis session and to study the long-term effect of binding competitor infusion during dialysis, we have developed a mathematical model that describes the intra- and inter-dialytic kinetics of PBUTs. The baseline model, without a binding competitor, was calibrated and validated using clinical data [15]. The model captures the dynamic equilibrium between PBUT and protein, as well as competitor drug and protein. In silico analysis informed us that competitive binding during HD significantly outperforms other state-of-the-art dialysis therapies, namely pre- and post-dilution hemodiafiltration and ideal membrane adsorption (Figure 6) [22]. These in silico findings are qualitatively validated by Shi et al. 2021, who compared binding competition and adsorption (by albumin dialysis) in pre-clinical uremic rat models and observed that binding competition outperformed adsorption [20]. 

In Maheshwari et al., a binding competitor was infused in an arterial line at a constant rate during 4 h of simulated HD session. Adsorption was modeled such that any free toxin passing over to the dialysate side of the dialyzer membrane is completely adsorbed so that the toxin concentration on the dialysate side is always zero (analogous to an infinitely high dialysate flow rate). For a detailed description of the model and simulation setup, refer to the original article [22].

Model simulations suggest that strong binding affinity is not the only criterion for the choice of a good binding competitor. For example, a higher amount of TRP (2000 mg in 500 mL saline) can outperform ibuprofen (800 mg in 200 mL saline), even though tryptophan binding affinity to albumin is an order of magnitude lower than that of ibuprofen [16]. The ibuprofen dose was restricted to 800 mg in our simulations as per FDA guidelines for a single-dose administration. Prolonged (one-month) use of TRP reduces the IS and pCS time-averaged concentration by 28.1% and 29.9%, respectively, compared to conventional HD. In Figure 7, we highlight the long-term kinetics and time-averaged concentration of pCS with and without a binding competitor. Here, a typical HD subject 70 kg in weight was dialyzed 3 times a week × 4 h per session, and the binding competitor was infused upstream of a dialyzer at a constant rate during dialysis [22].

## 7. Treatment of Drug Intoxications

In further in silico analyses, we explored the use of binding competition beyond PBUT removal to improve the treatment of intoxication with protein-bound drugs using HD [23]. Specifically, we tested ibuprofen to treat carbamazepine intoxication and aspirin to treat phenytoin intoxication. In both scenarios, we observed a significantly faster lowering of drug concentrations with the use of binding competitors vs. standard dialysis. Our model also provided insights into the effect of binding competitor half-life; specifically, that the use of binding competitors with longer plasma half-lives further reduced the treatment time required to lower the concentration of the offending drug. As per Extracorporeal Treatments in Poisoning (EXTRIP) working group guidelines, extracorporeal therapy is only recommended in cases of severe intoxications with phenytoin or carbamazepine because the free drug fraction in less severe intoxications is low for dialysis to be effective [24,25]; however, binding competitor-augmented HD can be a treatment option even in less severe cases and may also outperform adsorption-based hemoperfusion [22]. It remains to be explored whether these predicted results hold true in clinical settings. A summary of all evidence is reported in Table 1 in chronological order of publication date.

Our mathematical model [22] can be used in a number of ways, e.g., (1) as a potential tool for pre-screening the efficacy of various binding competitor candidates in order to select the most promising ones for subsequent clinical studies; (2) to personalize the dose of a competitor drug based on patient size, PBUT levels, prescribed ultrafiltration volume, etc.; (3) to test other extracorporeal modalities such as albumin dialysis, hemodiafiltration, and hemoperfusion; (4) to test the efficacy of a combination of binding competitors where the competitor molecules may target different binding sites on the albumin molecule (as in Li et al. 2019 [18]); (5) to optimize the intra-dialytic infusion profile for the binding competitor(s) to maximize toxin removal for a given amount of binding competitor used; or (6) to compare binding competitor infusion before the start of dialysis (as used in the rat study by Shi et al. [19]) with infusion during dialysis (as used in the clinical study by Madero et al. [21]). Regarding the timing of binding competitor infusion, we believe that infusion into the patient before dialysis may have detrimental effects due to a temporary increase in systemic free PBUT concentrations. Similarly, in the treatment of drug intoxication, infusing the binding competitors before dialysis may increase acute toxicity and lead to adverse outcomes.

## 8. Discussion and Conclusions

Binding competition is a paradigm shift in extracorporeal renal replacement therapies. Research spanning from bench studies to first-in-man clinical studies suggest that binding competition during dialysis can significantly improve the removal of PBUTs. Computer simulations suggest that the reduction ratio of strongly bound PBUTs can be improved from 35% in conventional HD to 60% in binding competitor-augmented HD [22]. Note that removal depends on the choice [17] and infused amount [22] of binding competitor.

Other extracorporeal techniques to improve PBUT removal are also under development. Borrowing from kidney physiology, Jansen and co-workers impregnated the blood-side of hollow fibers with organic anion transporter-1 [26]. In in vitro studies, this novel bio-membrane significantly reduced PBUT levels; results from pre-clinical studies are to follow. Though effective, the development of such bioengineered kidney tubules is complex and may be cost prohibitive. A new class of medium cut-off (MCO) membranes may leak a significant amount of protein—in a crossover study, use of a MCO dialyzer resulted in a 0.45 g/dL reduction in median albumin concentration in 3-month period [27]. Theoretically, this albumin loss can augment PBUT removal. Non-extracorporeal interventions to lower PBUTs level include dietary protein restriction, biotic supplements, or use of oral adsorbents such as AST-120. These techniques primarily focus on reduced production or reduced absorption of PBUTs in the gut [28].

Though effective and attractive, the binding competition approach raises some important questions, e.g., regarding the accumulation of binding competitors in kidney failure patients with long-term use. Another consideration is the risk-to-benefit ratio of chronic use of binding competitors. In other words., chronic ibuprofen use may accelerate the loss of residual renal function and may cause gastrointestinal bleeding [29]. Ideally, binding competitors with minimal side effects and, if possible, even salutary effects, would be chosen. Free fatty acids and tryptophan may be viable candidates. However, they compete only for one albumin binding site (the one where IS and pCS bind). To remove other PBUTs such as CMPF and HA, one or more additional binding competitors need to be infused. More research is required before the binding competition approach may be used in clinical practice. Such research should focus on the following: (1) the identification of ideal binding competitor candidates that target important PBUTs and have a favorable risk profile; (2) studying the short- and long-term effects associated with the use of these binding competitor(s) on pre-dialysis PBUT concentrations as well as on patient outcomes. Provided that such ideal binding competitors can be identified and demonstrated to have a net-positive effect, their application in routine HD would likely be technically simple and relatively inexpensive. In summary, binding competitor-enhanced dialysis holds promise for significantly improving PBUT removal compared to current extracorporeal renal replacement therapies. Furthermore, the application of binding competition holds promise for rendering HD a viable therapy option in the treatment of intoxications with highly protein-bound drugs.

## Figures and Tables

**Figure 1 toxins-13-00622-f001:**
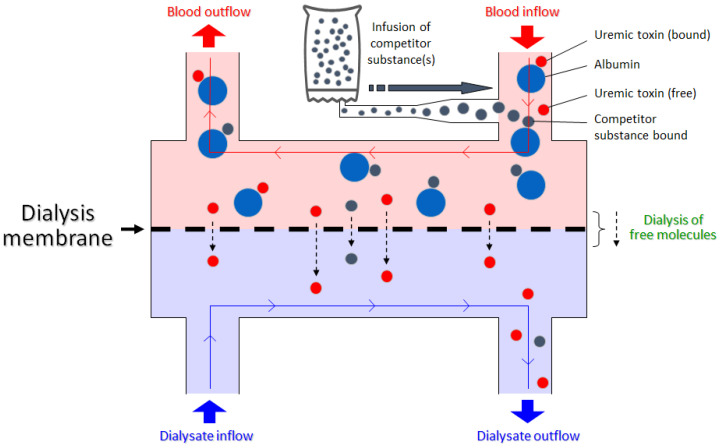
Schematic of binding competition between the protein-bound uremic toxin and competitor substance for the same binding site. The competitor drug is infused pre-dialyzer, leading to an increase in free toxin concentration and thus improved dialytic removal.

**Figure 2 toxins-13-00622-f002:**
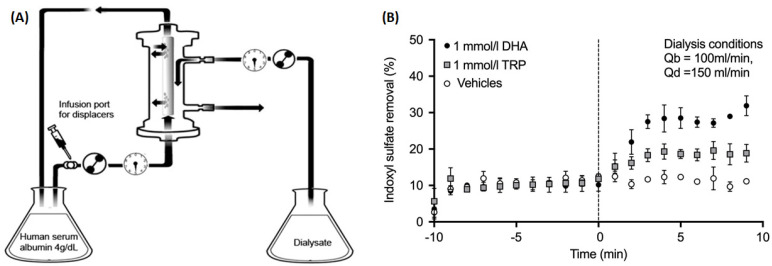
(**A**) In vitro dialysis setup where albumin solution spiked with indoxyl sulfate was dialyzed against standard dialysate solution. Two binding competitors (denoted as “displacers”) were tested individually. (**B**) Indoxyl sulfate removal (measured at the dialysate outlet) for 10 min before and 10 min after starting the infusion upstream of the dialyzer (start of infusion denoted by the vertical dashed line). Three types of infusions were tested: phosphate-buffered saline (PBS) only (“Vehicle”), PBS with 1 mmol/L tryptophan (“TRP”), and PBS with 1 mmol/L docosahexaenoic acid (“DHA”).

**Figure 3 toxins-13-00622-f003:**
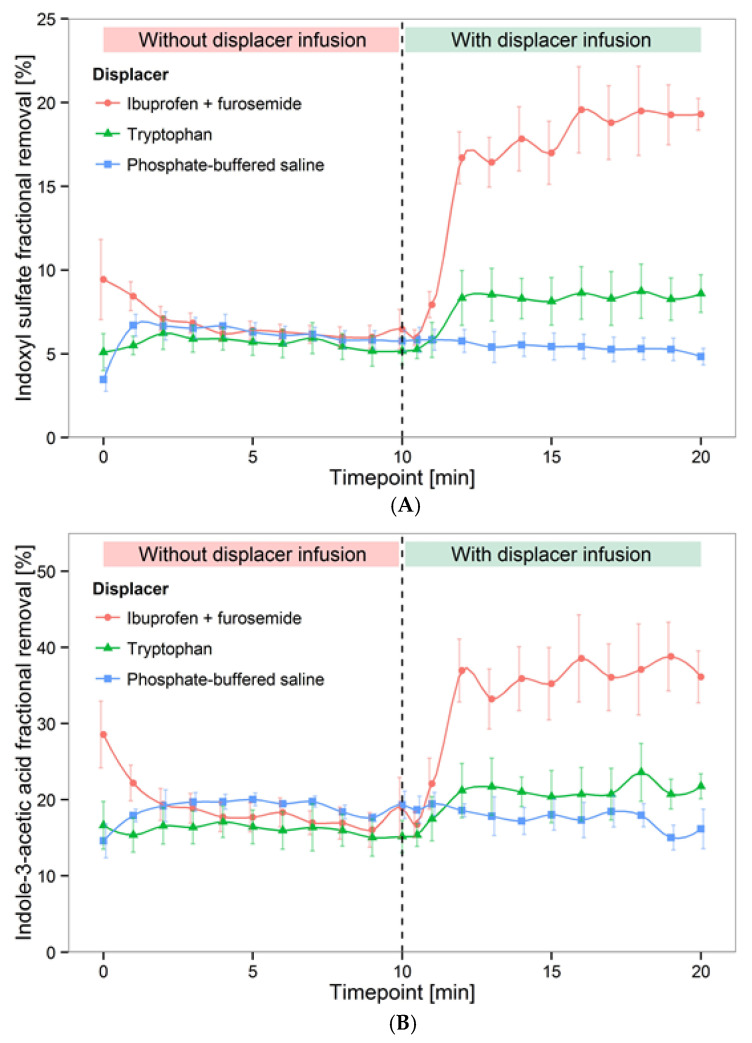
Binding competitor (denoted as “displacer”) infusion in ex vivo setup improved fractional removal of (**A**) indoxyl sulfate, (**B**) indole-3-acetic acid, and (**C**) hippuric acid. In this bench setup, uremic blood was dialyzed conventionally for the first 10 min, followed by infusion of a binding competitor for 10 min.

**Figure 4 toxins-13-00622-f004:**
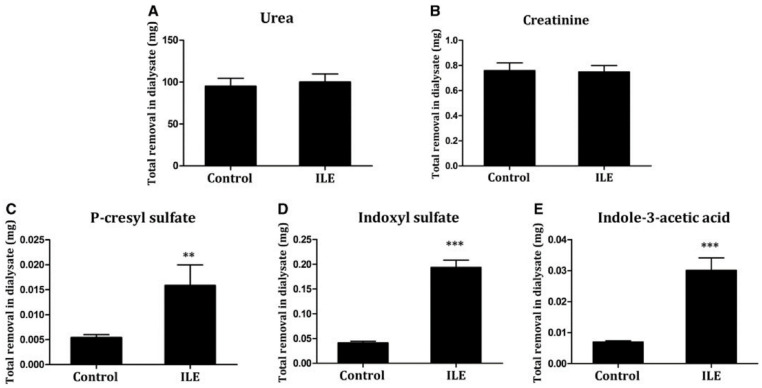
Total removal of (**A**) urea, (**B**) creatinine, (**C**) p-cresyl sulfate, (**D**) indoxyl sulfate, and (**E**) indole-3-acetic acid in 3 h dialysis in pre-clinical 5/6 nephrectomized rat model. Removal was studied in the control arm vs. intralipid emulsion infusion (binding competitor) arm (figure obtained with permission from NDT). *** *p* < 0.001 and ** *p* < 0.01 compared with the control group.

**Figure 5 toxins-13-00622-f005:**
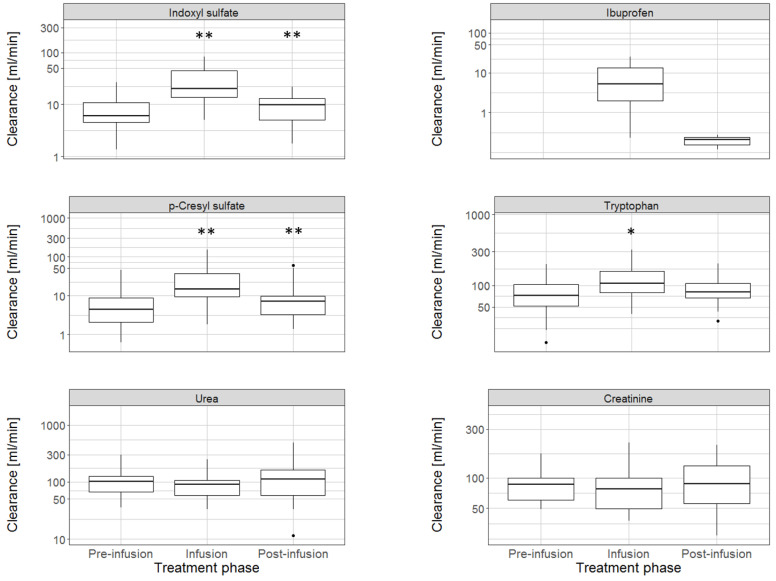
Dialytic clearance of uremic solutes during conventional hemodialysis before, during, and after infusion of a binding competitor into the arterial line upstream of the dialyzer. Ibuprofen was used as the binding competitor. Compared to the pre-infusion phase, there was a significant increase in indoxyl sulfate and p-cresyl sulfate clearance during the ibuprofen infusion, while the clearance of urea and creatinine (non-protein bound solutes) did not change. [* *p*-value, 0.01 compared with preceding phase; ** *p*-value, 0.001 compared with preceding phase; based on Wilcoxon signed rank test].

**Figure 6 toxins-13-00622-f006:**
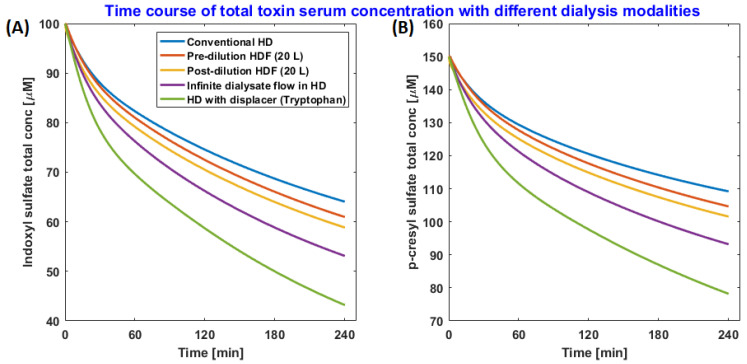
Time course of (**A**) total indoxyl sulfate (IS) and (**B**) p-cresyl sulfate (pCS) serum concentration with different extracorporeal dialysis modalities. The line color legend shown in the left panel applies to both plots.

**Figure 7 toxins-13-00622-f007:**
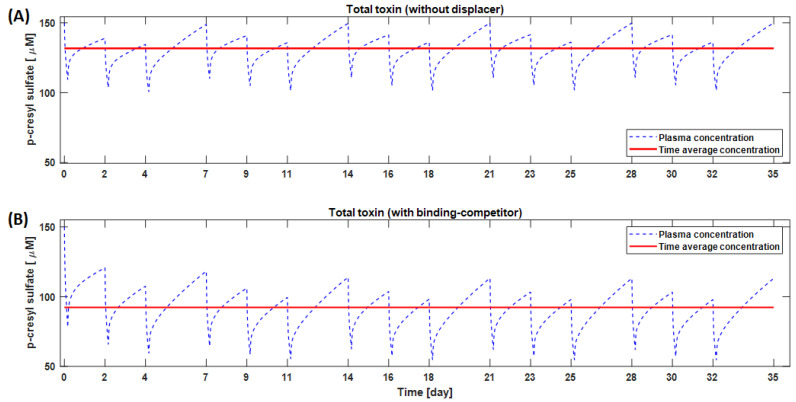
Monthly time-course of p-cresyl sulfate (pCS) concentration without (**A**) and with binding competitor (**B**). The binding competitor tested in these simulations was 2000 mg of tryptophan dissolved in 500 mL saline.

**Table 1 toxins-13-00622-t001:** Summary of existing evidence regarding use of binding competitor(s) for removal of protein-bound uremic toxins (PBUTs).

Study Reference	Study Setting	PBUT(s) Studied	Binding Competitor(s) Used	Study Metric	Study Outcome
Tao et al. 2015 [13]	In vitro	IS	TRP or docosahexaenoic acid (DHA) infused in extracorporeal circuit at constant rate	Fractional removal in the dialysate	TRP improved IS fractional removal from 10.2% at baseline to 18.5%; DHA improved the IS removal to 27.7%
Tao et al. 2016 [17]	Ex vivo	IS, IAA, HA	Ibuprofen + furosemide or TRP infused in extracorporeal circuit at constant rate	Fractional removal in the dialysate	Ibuprofen + furosemide improved IS removal from 6.4% to 18.3% and IAA removal from 16.8% to 34.5%; TRP improved IS and IAA removal to 10.5% and 27.1%, respectively.
Li et al. 2019 [18]	Pre-clinical uremic rat model	IS, pCS	Danhong injection or lithospermic acid infused intravenously at constant rate during latter 2 h of 4-h microdialysis.	Removal in first 2 h (without infusion) vs. latter 2 h (with infusion)	IS and pCS removal in dialysate improved by 119.5% and 127.6%, by lithospermic acid, respectively, which made up of 88% and 47%, respectively, of the total displacement effects of IS and pCS introduced by Danhong injection.
Maheshwari et al. 2019 [22]	In silico analysis of IS and pCS removal during HD	IS, pCS	TRP or ibuprofen infused into the extracorporeal circuit at constant rate during 4-h HD	Time-averaged concentration (TAC) after 1 month	TRP infusion in every HD session reduced the TAC by 28% for IS and 30% for pCS.
Shi et al. 2019 [19]	In vitro	CMPF, IAA, IS, pCS	Free fatty acids infused in extracorporeal circuit at constant rate	Fractional removal in the dialysate	CMPF fractional removal improved to 14.4% vs. no removal at baseline; pCS, IS, and IAA fractional removal from 8%, 11.7%, and 15.7% at baseline to 28%, 35%, and 40%, respectively.
Shi et al. 2019 [19]	Pre-clinical uremic rat model	pCS, IS, IAA	Intralipid™ (20%) infused intravenously 30 min before start of dialysis	Total solute removal in spent dialysate	Removal of pCS, IS, and IAA increased approximately 300%, compared to control.
Madero et al. 2019 [21]	First-in-man proof-of-concept study in 18 ESKD patients on maintenance hemodialysis	IS, pCS, HA, TRP	Ibuprofen infused at constant rate during 20–40 min of 4-h HD	Dialysate clearance comparison during pre-infusion phase (0–20 min) vs. infusion phase (21–40 min)	Clearance improved from 6.6 mL/min to 20 mL/min for IS, and 4.4 to 14.9 mL/min for pCS; TRP clearance increased moderately. Urea and creatinine clearance were unchanged.
Maheshwari et al. 2020 [23]	In silico analysis of drug intoxication treatment	Phenytoin, Carbamazepine	Infusion in extracorporeal circuit at constant rate. For phenytoin, aspirin was infused; for carbamazepine, ibuprofen was infused	Time required to bring patient back into therapeutic concentration range	For phenytoin, constant aspirin infusion reduced the HD time from 460 min to 330 min; for carbamazepine, constant ibuprofen infusion reduced the HD time from 265 min 220 min.
Shi et al. 2021 [20]	Pre-clinical uremic rat model	IS, pCS, IAA, HA	Intralipid™ infused intravenously 30 min before start of dialysis; albumin dialysis with bovine serum albumin; Combination of binding competition and albumin dialysis	Total solute removal in spent dialysate	In the Intralipid™ arm, approximately 10-fold increase in IS and pCS removal compared to control arm.

CMPF: 3-carboxy-4-methyl-5-propyl-2-furanpropionic acid; ESKD: end stage kidney disease; HA: hippuric acid; IAA: indole-3-acetic acid; IS: indoxyl sulfate; pCS: p-cresyl sulfate; TRP: tryptophane.

## Data Availability

All evidence presented in this manuscript exist in public domain.

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
