# Peer review of "Removal of Protein-Bound Uremic Toxins Using Binding Competitors in Hemodialysis: A Narrative Review"

_toxins, 2021, doi:10.3390/toxins13090622_

Round 1
Reviewer 1 Report
This is well written manuscript. The manuscript reviewed in vitro, ex vivo, in vivo and in silico removal of protein-bound uremic toxins using binding competitors during hemodialysis. In vitro and ex vivo evidence looks promising, but only one in vivo study in humans showing the potential of less toxic competitors to remove toxic compounds during dialysis. As suggested by the authors, more research is needed to identify effective competitors; and the studies should assess the identified competitors' safety profile. Additionally, more experimental studies in appropriate animal models are needed to assess the efficacy of the identified compounds followed by few clinical trials to bring these compounds to clinical practice.
Author Response
We thank the reviewer for highlighting the lack of sufficient in vivo studies. Unfortunately, there is only 1 in vivo study as we write. We hope that this review will motivate clinical researchers to carry out more in vivo research for PBUTs removal using binding-competitors.
Reviewer 2 Report
Good and timely article on an important issue. The article is overall very well written and I think it will be very useful for scientific community. Well done!
Author Response
We thank the reviewer for considering our article for publication.
Reviewer 3 Report
This manuscript aims to review the studies of a binding-competitor-enhanced dialysis method that had been tested in silico, ex vivo, and in vivo that proved to be an effective dialysis method. The previous studies demonstrated significantly improving protein-bound uremic toxins (PBUT) removal compared to the other extracorporeal renal replacement therapies. In addition, to review the application of a binding-competition dialysis method that might be a viable therapy option in the treatment of intoxications with highly protein-bound drugs. However, some of main important issues need to be verified to improve your work as following.
- Systematic reviews and meta-analyses are considered to be the highest quality evidence on a research topic because their study design reduces bias and produces more reliable findings. Please add evidence from recent systematic review and meta-analysis.
- Please summarized the results of previous studies, instead of present exact number and statistics for avoiding plagiarism.
- Over the last few years, the development of more advanced dialysis systems to improve the removal of protein-bound uremic toxins, or the application of binding competitors such as ibuprofen, folates and charcoal sorbents. Please provide evidence of folates and charcoal sorbents. (Vanholder R, Argilés A, Jankowski J. A history of uraemic toxicity and of the European Uraemic Toxin Work Group (EUTox). Clinical Kidney Journal. 2021 Jun;14(6):1514-23.)
4. Finally, since I am not a native English user, I did not check for grammatical errors thoroughly. This should be done by an appropriate language reviewer.
Author Response
We sincerely thank the reviewer for the thorough review of our work and emphasizing important research work that should be highlighted in our review. Please see below our point-by-point responses:
- We completely agree with the reviewer’s assessment that systematic reviews and meta-analyses provide evidence of the highest quality. However, studies pertaining to PBUTs removal using binding-competitor are heterogenous in nature (in vitro, ex vivo, and in vivo). The binding-competition method is still far from clinical reality; we have knowledge of one clinical study (Madero et al., 2019). We are not aware of systematic reviews or meta-analyses regarding the clinical use of PBUT displacers. In the light of the reviewer’s comment, we decided to slightly modify the manuscript title to “Removal of protein-bound uremic toxins using binding-competitors in hemodialysis: A narrative review.” Using the word “narrative” will help to avoid confusion with a systematic review.
We have added a reference (Takkavatakarn et al., 2021) that provides a systematic review of non-extracorporeal strategies for lowering PBUTs levels.
- Thank you for highlighting this important point regarding plagiarism. It goes without saying that we adhere to established standards: we have obtained explicit permission to use from Shi et al 2019 and reference the sources; we did not modify figure (Figure 4 in revised manuscript). At the same time, we have refrained from verbatim referencing results from highlighted studies and have used numeric values/results from previous research instead. Remaining figures are from our own publications.
- We did scrutinize the article by Vanholder et al., 2021 [Vanholder R, Argilés A, Jankowski J. A history of uraemic toxicity and of the European Uraemic Toxin Work Group (EUTox). Clinical Kidney Journal. 2021 Jun;14(6):1514-23]. Despite careful exploration, we could not locate any literature reference on folate as binding-competitor to improve PBUTs removal. To the best of our knowledge, such literature is referenced neither in the Vanholder et al. 2021 paper nor in other public sources. We acknowledge that we could be wrong; if you could provide such a reference, we will gladly include all pertinent details in the revised manuscript.
- Thank you; we conducted a thorough language review.
Round 2
Reviewer 3 Report
The authors addressed all my previous concerns and significantly improved quality of the manuscript. I have no additional comment.